# Human Chondrocytes, Metabolism of Articular Cartilage, and Strategies for Application to Tissue Engineering

**DOI:** 10.3390/ijms242317096

**Published:** 2023-12-04

**Authors:** Darina Bačenková, Marianna Trebuňová, Jana Demeterová, Jozef Živčák

**Affiliations:** Department of Biomedical Engineering and Measurement, Faculty of Mechanical Engineering, Technical University of Košice, Letná 9, 042 00 Košice, Slovakia; marianna.trebunova@tuke.sk (M.T.); jana.demeterova@tuke.sk (J.D.); jozef.zivcak@tuke.sk (J.Ž.)

**Keywords:** chondrocyte, mesenchymal cells, collagen type II, articular cartilage, integrins

## Abstract

Hyaline cartilage, which is characterized by the absence of vascularization and innervation, has minimal self-repair potential in case of damage and defect formation in the chondral layer. Chondrocytes are specialized cells that ensure the synthesis of extracellular matrix components, namely type II collagen and aggregen. On their surface, they express integrins CD44, α1β1, α3β1, α5β1, α10β1, αVβ1, αVβ3, and αVβ5, which are also collagen-binding components of the extracellular matrix. This article aims to contribute to solving the problem of the possible repair of chondral defects through unique methods of tissue engineering, as well as the process of pathological events in articular cartilage. In vitro cell culture models used for hyaline cartilage repair could bring about advanced possibilities. Currently, there are several variants of the combination of natural and synthetic polymers and chondrocytes. In a three-dimensional environment, chondrocytes retain their production capacity. In the case of mesenchymal stromal cells, their favorable ability is to differentiate into a chondrogenic lineage in a three-dimensional culture.

## 1. Introduction

Articular cartilage is a load-bearing connective tissue that has a low self-repair potential. There are high demands placed on articular hyaline cartilage in the organism, mainly mechanical flexibility, load-bearing capacity, and the ability to reduce friction. The function of the cartilage in joints is to ensure low friction and the ability to distribute the weight load acting in the joint. An articular cartilage defect can persist without healing, or if it extends into the blood-filled subchondrium, then it is replaced by cartilage tissue that does not have suitable strength properties [1,2,3]. Clinical orthopedics has long been devoted to the problem of repairing articular cartilage defects in diarthrodial joints. Currently, there are several recommended procedures, but an optimal repair procedure is not available. For an ideal solution, and to choose a correct reparative procedure, it is necessary to know in detail the pathological processes that affect relationships and processes at the molecular and cellular levels in the cartilage. Articular cartilage contains chondrocytes as a single cell type, existing without a blood supply or the presence of innervation. Due to its simpler cartilage construction, it was considered a suitable model for tissue simulation in vitro [4]. This review is devoted to the basic biochemical processes in hyaline cartilage based on structural amino acids and current cell populations. The overview is also dedicated to the possibilities of in vitro culture of chondrocytes and detailed descriptions of current methods for three-dimensional (3D) cultivation.

## 2. Articular Cartilage

Hyaline cartilage is composed of a complex structured extracellular matrix (ECM) that facilitates friction on the surface of the articular cartilage and is adapted to repeated compression during movement. Synovial fluid has an important function in the nutrition of articular chondrocytes, as this process is ensured by diffusion and improving the lubrication of cartilage joint surfaces [5]. The joint capsule with its cell lining on the surface of the synovial membrane contains cells of the synovial intima called synoviocytes. Fibroblast-like cells are involved in the production of synovial fluid, and the macrophage-like type can be considered resident macrophage cell types [6]. In the organism, articular cartilage is located in the articular surfaces at the end of the epiphyses of long bones and consists of several interconnected zones, namely superficial, transitional, deep, and calcified areas separated from the underlying bone. Parts of the pelvis and the bones of the long limbs are replaced by bone through the process of ossification [7].

### 2.1. Extracellular Matrix

Hyaline cartilage contains 70–80% water from the total volume, and the main components of the ECM are collagen and proteoglycans, comprising up to 50–70% of the dry weight. Specific cartilage is characterized by the presence of collagen type II (COL2) and the proteoglycan aggrecan. It is composed of layers differing in collagen type composition, relative orientation of collagen fibers, and cell density and cytomorphology, which generates a hierarchically organized cartilage structure. Chondrocytes, which represent a low volume percentage of the total tissue share, do not have a direct mechanical role. These cells in the ECM are without detectable mitotic activity in cartilage with a low metabolic turnover. The main functions of chondrocytes are the synthesis and degradation of the ECM [8]. The metabolic activity of chondrocytes is focused on the synthesis and maintenance of the components of the ECM.

Hyaline cartilage has no innervation and no vascularization, and it is characterized by a low pH mainly in deep zones (~6.9), with a minimum pO_2_ of 2–5% [9,10]. The ECM of hyaline cartilage consists of a highly organized network of interconnected structural proteoglycans and collagens. The ECM of the articular joint creates conditions for the basic mechanical properties of cartilage. Reducing friction and lubrication is made possible by a complex of collagen fibers interacting with hyaluronic acid or hyaluronan (HA) and lubricin on the surface of the joint. Collagens act as a stabilizing element with the ability to resist tension, and proteoglycans, which are negatively charged and connected to the collagen network, attract Na^+^ and H_2_O cations. The process of binding water causes an increase in tension in the collagen network, which gives the tissue the power to resist compression [9,11]. In hyaline cartilage, type II collagen fibers are represented the most, and type I, IV, V, VI, IX, and XI collagen are also present but in smaller concentrations, and their role is to mutually strengthen the fibers [12].

In terms of size and content, glycosaminoglycans (GAGs) and aggrecans (ACANs) are significant, containing more than 100 chondroitin sulfate (CS) and keratan sulfate (KS) chains which are bound to the core protein. ACANs are reported to be the main proteoglycans of hyaline cartilage, which interact with HA and form complex proteoglycan aggregates [10]. ACANs are bound to the HA fiber by covalent bonding, while this interaction is stabilized by a termed link protein. An anionic charge is present on the individual molecules of ACANs containing sulfated GAGs chains, while their localization in the matrix ensures the formation of aggregates and is essential for the function of ACANs.

ACAN molecules in the form of proteoglycan aggregates with a high molecular weight mainly ensure the strength properties and increased mechanical load-bearing requirements in cartilage tissue [5]. Proteoglycans characterized by shorter and less structured chains than ACANs have the ability to interact with collagen molecules. Decorin, biglycan, and fibromodulin do have structural properties like proteins, but they differ in their function from GAGs. Both decorin and fibromodulin interact with COL2 fibrils in the matrix and may play a role in fibrillogenesis and interfibril interactions [13]. HA forms aggregate structures with ACANs that are stabilized by a binding protein. The resulting spatial network represents the structures that are anchored within collagen fibrillar networks in articular cartilage [14]. HA-ACAN aggregates affect the water content and influence the hydrostatic pressure in cartilage. In addition to the structural osmotic function, ACANs enable the transport of dissolved substances and essential nutrients in the tissue of hyaline cartilage, while this process is related to the ability of hydration. The structure and content of ACANs in cartilage are not constant during human life. Overall, it can be said that aggregate molecules provide cartilage with specific osmotic properties, thereby optimizing the ability of articular cartilage to withstand high-pressure loads. CS and KS form GAG, in which they act as structural molecules of extracellular matrices [15,16] (Figure 1).

### 2.2. Chondrocyte

The cell population of chondrocytes in healthy articular cartilage represents typical resting and differentiated cells, whose task is to maintain a dynamic relationship between anabolism and catabolism of the ECM [17]. Mature chondrocytes of hyaline cartilage are stored in lacunae in isogenetic groups of several cells in the territorial matrix located in hyaline cartilage in a specific physicochemical environment. The mentioned chondrocytes were created by mitotic division from one parent cell [3]. Chondrocytes have a round or oval shape and fill only 5–10% of the cartilage volume in total. The cytoplasm of chondrocytes is characterized by a small spherical nucleus, and it also contains the Golgi apparatus, mitochondria, and less numerous lipid droplets [18].

#### 2.2.1. Progenitor Cells of Cartilage

The process of cartilage formation is initiated in the early period of embryogenesis by a population of prechondrocytic mesenchymal stem cells (MSCs). Chondrogenesis is a process of condensation of MSCs by forming densely packed cell aggregates where, subsequently, differentiation into prechondrocytes occurs [19,20]. The ontogenesis of the organism includes the activity of growth processes in tissues, specifically the complex process of endochondral ossification. The initial differentiation of growth plate chondrocytes during endochondral bone in vivo is affected by signaling activated by homeobox genes and soluble mediators. The SOX gene family is bound by the amino acid sequences and DNA-binding domains of the high-mobility group box (HMGB). SRY-box transcription factor 9 (SOX-9) is present in the process, which has a regulatory role in post-transcriptional and post-translational transcripts. Zhao et al. described cooperation between the high expression of SOX-9 and the high expression of collagen type II alpha-1 gene (Col2A1) with the assumption that it is necessary for the full expression of the chondrocyte phenotype. The Col2A1 gene encodes the COL2 molecule, which is characteristic of hyaline cartilage [21]. The differentiation and proliferation of chondrocytes are influenced by growth factors and, at the same time, interaction through contact with the ECM, which is essential for the maintenance of the cell phenotype. During the differentiation of chondrocytes, the cells and the microenvironment of the ECM cooperate through integrin receptors, which are part of the ECM [22].

#### 2.2.2. Characteristic Phenotype of Chondrocytes

Mature chondrocytes produce structural proteins, COL2, collagen types IX (COL9) and XI (COL11), and ACAN [23]. Articular chondrocytes with the unaffected, natural phenotype also synthesize lubricin, or proteoglycan-4 (PRG4), and glycoprotein. Its lubrication function is significant and has an effect on the reduction of friction between the applied cartilage surfaces. PRG4 has a suppressive effect on the function of inflammatory cytokines and their activation of the proliferation of RA synovial fibroblasts [24].

The integrin family has a role in the process of cell adhesion as well as in cell-to-cell interactions and interactions with the ECM. Integrins as transmembrane receptors recognize specific ECM molecules [3,19,25]. The intracellular part of integrins, extending into the cytoplasm, is connected to the cytoskeleton of the cell. The extracellular part is connected to ligands and triggers cell activation. Signaling through integrins enables transmitting signal molecules in both directions. Articular chondrocytes express α1β1, α3β1, α5β1, α10β1, αVβ1, αVβ3, and αVβ5 integrins [22,26]. The α5β1 and αVβ3 integrins bind to the sequence Arg-Gly-Asp (RGD), which is contained in several ECM proteins [22].

Integrin α10β1 is an important integrin that binds collagen in cartilage tissue. Camper et al. described integrin α10β1 as a type II collagen-binding receptor. It is a unique marker for determining the phenotype of chondrocytes. Integrin α10β1 is part of the cell–matrix interaction which is essential for cartilage development and the chondrogenesis of MSCs [26,27]. The function in the morphogenesis of growth plates was observed by the authors in an animal model with a disorder, namely a deletion for the alpha10 integrin gene. A defect in the ontogenesis and growth of long bones were manifested in mice [28]. Furthermore, chondrocytes express types of integrins binding to ECM molecules [29]. Achorin V belongs to annexins, being a molecule that binds to type II collagen. The most abundant receptors for fibronectin (α5β1) are for COL2 and VI (α1β1, α2β1, and α11β1), vitronectin, osteopontin (αVβ3), and laminin (α6β1) [30,31]. An important membrane receptor of chondrocytes is the hyaluronic receptor CD44 [32]. The binding of chondrocytes and HA affects the homeostasis of the cartilage environment [33]. In the case of blocking the binding of CD44 and HA, damage to and degradation of the ECM occurs [29].

#### 2.2.3. Specific Growth Factors during the Expansion of Articular Chondrocytes

In addition to the binding of integrins to ECM molecules, cell metabolism is also regulated in a paracrine influence (i.e., by the release of soluble factors). Several studies have reported that growth factors have a stimulating effect on the proliferation of mammalian chondrocytes [34,35,36]. The transforming growth factor beta (TGF-β) superfamily has a broad role in physiological and pathological events, affecting the adhesion, growth, and differentiation of a variety of tissue cells [37]. Over 30 proteins belong to this group of activins and bone morphogenetic proteins (BMPs) [38]. It is involved in maintaining homeostasis, embryogenesis, and immune events [39,40]. Regarding biochemical events in connective tissues, TGF-β participates in the formation of cartilage and bone tissue [41]. TGF-β activates signaling cascades, of which the TGF-β/Smad signal pathway is the best known one, for enacting and modulating the gene expression of several proteins [42]. After activation with other factors, TGF-β is able to form a serine/threonine kinase complex [43]. This triggers the signaling pathways that regulate differentiation, proliferation, and immune processes [44,45]. It is an interesting fact that animal models of rodents that have excessive expression of TβRII in cartilage are often affected by joint arthrosis [46]. Furthermore, they have an increased expression of markers for collagen type X (COL10) [47].

The most widely studied growth factor from the TGF-β superfamily is the TGF-β1 factor [48]. TGF-β2 is also a subject of studies on disorders and the treatment of cartilage defects [49]. TGF-β contains the following homologous dimeric isoforms: TGF-β1 and TGF-β3. TGF-β has a downregulating effect on osteogenesis, while its pleiotropic function is dependent on the interaction with the surrounding environment. TGF-β2 is involved in the Smad canonical signaling pathway and is also involved in generic mitogen-activated protein kinase (MAPK) activation in chondrocytes [50]. A possible function of TGF-β2 in chondrocyte redifferentiation through activin receptor kinase 5 (ALK5)/Smad3 under hypoxic conditions was considered [51]. In the experiment, the authors observed the redifferentiation of chondrocytes in culture with the stimulating effect of the growth factors, namely fibroblast growth factor (FGF) and TGFβ [52]. Due to the influence of the mentioned growth factors, the chondrocytes showed an increased proliferation rate compared with the control, as well as differentiation into a chondrocyte phenotype with COL2A1 expression [35].

BMPs belong to the TGF-β family. BMPs have a stimulating effect during chondrogenesis. They preferentially support the condensation step at the beginning of chondrogenesis. The process is related to the stimulation of N-cadherin expression promoting cell-cell contact. BMPs affect the expression of SOX-9 and type II collagen [18].

#### 2.2.4. Biomechanical Principles of Articular Cartilage and Chondrocytes

Movement is an essential part of the life of living organisms. At the cellular level, complex mechanical stress acts on tissues and cells. Cartilage metabolism during ontogeny and throughout the life of organisms is influenced by mechanical factors that directly control the activation and expression of genes for growth, metabolism, and the resulting phenotype of chondrocytes [53]. Articular chondrocytes respond to mechanical signals transmitted through the ECM with metabolic activity. Mechanical influences such as intermittent fluid pressure act to preserve the chondrocyte phenotype. Changes in the shape and volume of chondrocyte nuclei in connection with ECM deformation were discovered. During micromechanical analysis, chondrocytes were observed to have viscoelastic properties but with lower mechanical strength than pericellular ECM [54]. Mild tension and shear action have a stimulating effect on growth and ossification. During ontogenesis, the thickness of the articular cartilage is formed, while it is most massive in the place with the greatest bearing load. In articular joints, the health of the cartilage is proportional to its load [55]. The production of ECM by chondrocytes depends on the zone in which they are located. Superficial zone chondrocytes exposed to fluid flow and mechanical pressure produce higher amounts of type II collagen compared with chondrocytes in the middle and deep radial zones, which synthesize high amounts of GAGs in addition to type II collagen [55,56].

From a mechanical point of view, cartilage has a multiphase nature. Cartilage can be characterized from a physicochemical point of view as a viscoelastic material that contains the following two-phase environment: a solid phase and a fluid phase [57]. The solid phase contains ECM components and collagen fibers linked to ACAN and HA. The liquid phase consists of water with a content of up to 80% wet weight and ions, as well as calcium, sodium, and chloride. Mechanical loading of joints is essential for the nutrition of articular chondrocytes and the synthesis of ECM components. Immobilization and restriction of locomotion causes a decrease in the metabolic activity of chondrocytes and a functional weakening of the cartilage [58]. The experimental results showed that up to 75% of the applied load on the joint surface was compensated by the liquid phase. The viscoelastic properties of the articular cartilage ensure the functionality of the joint under repeated loading. When articular contact forces arise during loading, the pressure of the interstitial fluid increases. When the joint is loaded, the pressure increases. The ECM environment is permeable, and with increased pressure, part of the fluid from the ECM can pass into the liquid phase. After the pressure is removed, the interstitial fluid flows back. The process of multi-phase layers in the articular cartilage ensures low friction and the resulting biomechanical functionality of the articular joints.

## 3. Degenerative Processes Affecting the Metabolism of Cartilage

The processes caused by the aging of articular cartilage are correlated with the presence of risk factors such as oxidative stress and significant mechanical loading. The aging of chondrocytes in articular cartilage compared with the processes during osteoarthritis (OA) are different processes. Paradoxically, for several points, the mentioned biological processes are extremely similar. With increasing age, the prevalence of OA in the population increases significantly.

### 3.1. Age-Related Degeneration of Articular Cartilage

It is generally assumed that processes affected by age are conditioned by cell replication disorders, which are related to cartilage metabolism, and homeostasis resulting from the relationship between anabolism and catabolism [59]. Changes in the structure of stressed tissue are conditioned by processes in the cell and intracellularly related to GAG synthesis [60]. The process involving extracellular effects is mainly related to nuclear protein and HA degradation [61]. During life, especially during development and growth, the GAG chain changes. The most significant change is the shortening of the CS length, while the sulfation also changes from position 4 of N-acetylgalactosamine to position 6 [10,62]. Chain KS lengthening was observed. In context with increasing age, the heterogeneity of ACANs, which are the structural organizers of the ECM, has been noticed [9,63]. ECM-based injectable thermo-sensitive hydrogel is used in the recovery of injured cartilage induced by OA.

The significant impact of oxidative stress and reactive oxygen was the impetus for the beginning of the study of the pathological process of glycation (i.e., protein modification) [64]. Advanced glycation end products (AGEs) are known as heterogeneous molecules derived from the non-enzymatic reaction products of glucose and other sugar derivatives with proteins. AGEs are one of the newer biomarkers for the aging process and are observed in several disease states [65]. These molecules are formed during glucose metabolism and through glucose degradation, as well as Maillard reactions [66]. Maillard reactions in vivo represent the pathways of chemical protein modifications from carbohydrates [67]. The process of glycation (i.e., protein modification) contributes to the aging of body proteins and also has a regulatory function in physiological reactions, as well as pathological processes [68]. The endogenous production of AGEs in the organism is also influenced by environmental agents. The generation of AGEs is an irreversible process. Glycated molecules accumulate in tissues [69]. As a result, structural and functional pathological changes occur in tissues. Molecules associated with AGEs are often present in musculoskeletal disorders [70,71].

### 3.2. Oxidative Stress

Mitochondria are the metabolic center of the eukaryotic cell. Energy exchange is carried out through the tricarboxylic acid cycle, also called the Krebs cycle, and oxidative phosphorylation [72]. Adenosine triphosphate (ATP) synthesis takes place here. The production and regulation of reactive oxygen species (ROS) occurs in the mitochondria, and this process is related to hypoxia-inducible factor-1 (HIF-1) [73]. In cellular metabolism, ROS are a physiological product that have a function in cell signaling. The term oxygen toxicity was used for the adverse effect of ROS [74]. The presence of an unpaired electron in their outer shell creates instability and their potential reactivity [75]. In animal cells, electron transfer takes place in mitochondrial organelles. Redundant electrons with O_2_ create superoxides. Therefore, the ROS levels are minimal. The wider group of oxygen radicals includes superoxide anion radical, hydroperoxyl radical, hydroxyl radical, hydrogen peroxide, lipid peroxyl radical, urate radical, α-tocopherol radical, and ascorbate radical [76]. In case of an imbalance of production and homeostasis of ROS, there is a risk of modification of sensitive cellular structures, specifically damaged DNA [77]. The negative impact of ROS can arise from their increased production in the presence of stress factors in the organism. Antioxidants, mainly ascorbic acid (AA) and beta-carotene, are active at this point [78]. Accordingly, oxidative stress is one of the factors in the development of OA [79]. It is known from the literature that the production of free radicals was increased, and the level of antioxidant forces was decreased in the main OA patients. Oxidative stress is considered an important part of the articular cartilage degradation process and is directly related to aging as well as OA. Chondrocytes affected by the negative influence of the environment with the presence of ROS are subject to a heavy load and cellular stress [80]. AA may be a potential drug for therapeutic intervention and to slow the progression of age-associated diseases [81].

## 4. Diseases of Articular Cartilage

Articular cartilage defects are most often caused by direct mechanical trauma. Depending on the depth, defects are classified as chondral or osteochondral [82,83]. Repairing articular cartilage defects at an early stage can delay the onset and later progression of OA [84]. Genetically determined diseases of the cartilage mainly include autosomal dominant diseases, such as a form of dwarfism called achondroplasia, which is caused by dominant point mutations in the transmembrane domain of FGF receptor 3 (FGFR3). There are other forms of FGFR3 mutation, such as hypochondroplasia and thanatotropic dysplasia, which is often fatal. In the case of mutations that relate to components of the extracellular matrix, they are in the Col2A1 gene for type II collagen. This is a broader spectrum of disorders called type II collagenopathy. Diseases with collagen disorders cause abnormal growth and earlier onset of arthrosis. Osteochondritis dissecans is a disease in which loose bodies form in the joints without previous trauma. It causes a focal, idiopathically altered subchondrium with an adjacent defective part of the articular cartilage. The etiology of the disease is multifactorial and not fully clarified. Frequent inflammations and repeated minor traumas, ischemia, and hereditary factors are mentioned as the main triggers of this disease. The most frequently affected area is the knee joint [16]. With increasing age, the prevalence of OA increases significantly with the changes that occur in the articular cartilage. Degeneration of articular cartilage is the result of aging and excessive mechanical stress, especially if it is associated with joint instability.

### 4.1. Osteoarthritis

Osteoarthritis is a very common joint disease that begins with the degeneration of joint cartilage, with pathological changes in the cartilage and the bone tissue. In the case of pathological processes, the production of collagen by chondrocytes changes.

In OA, the cell phenotype changes and affects the chondrocytes’ expression of α2β1, α4β1, and α6β1, while the levels of α1β1 and α3β1 increase along with the appearance of α 2beta1, α 4beta1, and α 6beta1, which do not occur in healthy cartilage [26,85]. When the matrix is damaged during the process of OA, fibronectin fragments are formed, which activate pro-inflammatory and catabolic reactions through integrins. The damaging process continues and leads to ECM degradation. The phenotype of chondrocytes changes to hypertrophic and fibro-cartilaginous chondrocytes [86,87]. The altered phenotype may promote the development of the disease and the destruction of the articular cartilage. Under these specific non-physiological conditions, chondrocytes undergo hypertrophic differentiation, during which physiological production changes, and they produce COL10 to a greater extent, which is considered a marker of chondrocyte hypertrophy [88]. In the case of the transition of the chondrocyte phenotype from chondrotic to fibroblastic, there are changes in the structure of the cytoskeleton and an increase in the production of collagen type I (COL1) and decorin. Overall, this negatively affects the mechanical properties of the ECM and the formation of fibro-cartilage tissue [89]. In this situation, the chondrocyte phenotype is dedifferentiated. This process of chondrocyte phenotype instability occurs in the development of OA and is also probable in the repair of cartilage defects [90].

Chondrocyte hypertrophy occurs naturally during the development of endochondral ossification. Arnold et al. found that myocyte enhancer factor-2 (MEF2C), a transcription factor known to regulate the development of the muscular and cardiovascular systems, also corresponds to the development of the skeletal system by activating the gene program for chondrocyte hypertrophy. In contrast, the activated form of MEF2C acts on premature chondrocyte hypertrophy, or growth plate ossification [91]. MEF2C deletion impairs hypertrophy and angiogenesis during ontogenesis in rodent animal models. MEF2C has an effect on the regulation of Runt-related transcription factor 2 (RUNX2) and RUNX3, which are essential in the process of chondrocyte hypertrophy, and the process of osteogenesis. In contrast, RUN1 suppresses hypertrophic differentiation and promotes the production of the hyaline ECM [92]. This process is supported by SOX molecules that regulate chondrogenesis. Chondrocytes and their progenitors express SOX9, a transcription factor involved in chondrogenesis, and its cofactors SOX5 and SOX6. The said transcription factors stimulate the expression of COL2 and ACAN genes [8] (Table 1). SOX9 is a crucial transcription factor in the ontogenesis and metabolism of mature cartilage. SOX9 is involved in the regulation of cartilage development and acts stimulatingly as an activator of genes encoding regulatory factors and metabolic processes [93]. The characteristic phenotype of articular chondrocytes varies depending on the environment. Isolated chondrocytes in monolayer culture in vitro tend to dedifferentiate and produce COL1 through modification from COL2 [94]. Articular chondrocytes are characterized by the expression of glycoprotein and a tissue inhibitor of metallopeptidase protein 1 (TIMP1). TIMP1 is a member of the extended TIMP family. It is known as an inhibitor of protease enzymes that work by degrading the ECM. The transformation of the chondrocyte phenotype to hypertrophic and fibrocartilaginous is correlated with the development of OA [95,96].

Especially in the advanced grade of OA, chondrocytes lose their phenotype to hypertrophic and fibrocartilaginous chondrocytes, which participate in the catabolic processes of hyaline cartilage destruction. The change in markers on chondrocytes is related to dedifferentiation into fibrocartilaginous cells and hypertrophic chondrocytes [97]. The process of fibrosis has a role in the development of OA. This pathological process is manifested by the accumulation of connective tissue. Limited articular cartilage reparability is cell type-associated with chondrocyte dedifferentiation to a fibrosis-associated phenotype. The change in phenotype is manifested in the production of glycoproteins and proteoglycans, which are typical for fibrocartilaginous cartilage [98]. When re-expressing the chondrocyte phenotype, it is advisable to monitor the ratio for the expression of COL2/COL1 and SOX9/RUNX2 [99,100].

### 4.2. Rheumatoid Arthritis

Rheumatoid arthritis (RA) is a serious chronic and autoimmune disease that causes inflammation and limited joint function. In the case of RA, there is a significant dysregulation of immunity, which affects the metabolism of chondrocytes [24]. The altered chondrocytes in RA are used to contribute to the inflammation present in the rheumatoid joint. Several pro-inflammatory factors, mainly IL-1β, tumor necrosis factor (TNF)-α, IL-6, and IL-17, are involved in RA. Pro-inflammatory factors primarily activate catabolic processes that limit the functionality of chondrocytes. This can further initiate the process of apoptosis in the affected cells. Specifically, TNF-α inhibits chondrogenic differentiation, which is linked to p38 mitogen-activating protein kinase (MAPK) [24,101]. In addition to TNF-α, inflammatory cells also secrete interferon-γ (IFN-γ), which suppresses proliferation, and negatively affect the viability of chondrocytes. CD40 expression of articular chondrocytes was observed in RA, which conditions a massive cytokine response and the formation of matrix metalloproteinases (MMPs), which are enzymes of ECM degradation. In a pathological environment, chondrocytes release MMP-1, MMP-3, MMP-10, MMP-1, and MMP-13, which activate inflammation and enable angiogenesis. Degraded ECM fragments cause stimulation of the pro-inflammatory cytokines interleukins (IL)-6, IL-8, and monocyte chemoattractant protein (MCP)-1. In an inflammatory environment, toll-like receptor (TLR)-1, TLR-2, and TLR-4 can be expressed on chondrocytes. Subsequently, TLR-2 activation using IL-1, TNF-α, peptidoglycans, or fibronectin fragments affects and increases the production of MMPs and vascular endothelial growth factor (VEGF). During this process, nitric oxide (NO) is released, which acts as a strong factor of chondrocyte apoptosis and has a destructive effect in the arthritis process. The process of ECM destruction and the released degraded components of proteoglycans act as autoantigens for T-lymphocytes. The pathology of RA is a complex combination enhanced by cellular immunity and pro-inflammatory mediators [102].

## 5. Cartilage Repair Cell Types

The challenging task of tissue engineering is to prepare the most optimal complex of cells, scaffolds, and growth factors for the successful repair of chondral defects (Figure 2). Furthermore, in vitro culture methods of human and animal chondrocytes offer the possibility of simulating the physiological and pathological processes of articular cartilage. When inducing processes leading to catabolic states and cartilage degradation, it is possible to influence and subsequently monitor the parameters, chondrocyte phenotype, and production of active chemokines. Several types of cells are used to repair chondral defects, the most important ones being chondrocytes, which are approved for use by the US Food and Drug Administration (FDA) [103]. During ontogeny, as mentioned above, MSCs differentiate into chondrocytes [19]. For this reason, MSCs have also been widely tested as cells capable of differentiating into chondrocytes in vitro and applicable for cartilage tissue engineering [104].

### 5.1. Mesenchymal Stem Cells

Currently, MSCs are frequently used cells in the thematically broad field of tissue engineering [105]. MSCs participate in regeneration processes, angiogenesis, and differentiation in the organism. These cells were first described in the last century by Friedenstein et al. in rodent bone marrow culture [106]. The differentiation potential of MSCs has been demonstrated in numerous studies in vitro [107,108,109]. The following criteria are a condition for MSC classification according to the International Society for Cell Therapy (ISCT): the ability to adhere to the surface of a plastic culture; the proven expression of the surface antigens CD105, CD73, and CD90; no expression of the hematopoietic markers CD45, CD34, CD14 or CD11b, CD79a or CD19, and HLA class II; and the ability to differentiate into osteoblasts, adipocytes, or chondroblasts in vitro [107,110]. MSCs can be isolated from a number of sources, with cells originally being isolated from bone marrow. Currently, MSCs have been obtained from adipose tissue, synovial fluid, placental tissues, embryonic tissues, muscle, dental pulp, and liver [111,112,113,114]. MSCs have low immunogenicity in an allogeneic environment. This implies the possibility of using MSCs from an allogeneic source due to their low cellular immunogenicity [105,115,116]. The advantage of MSCs is their possible multiplication in vitro in an undifferentiated state, with preservation of the ability to differentiate by adding stimulating growth factors to the culture. The ability of MSCs to differentiate into a chondrogenic lineage is supported by pellet 3D culture under appropriate stimulation conditions [103].

### 5.2. Articular Chondrocytes

Chondrocytes for in vitro multiplication can be isolated from several sources of the organism, such as the non-load-bearing area of articular cartilage, nasal septum, costal cartilage, or ear cartilage [104]. Chondrocytes are a differentiated cell type in a resting state. They ensure the maintenance of the physiological state of the ECM of the articular cartilage with characteristic strength properties. In other words, the natural metabolism of chondrocytes in situ is adapted to low oxygen tension and no vascular supply. The cells are supplied with nutrients through the process of diffusion from the synovial fluid. These parameters influence changes in the expression of isolated and in vitro cultured chondrocytes, which have increased metabolic and proliferative activity [99]. Chondrocyte dedifferentiation in an organism causes degeneration of the intervertebral disc and is associated with OA, which is associated with cartilage damage. The dedifferentiation process of chondrocytes was also observed in vitro. In the process of cultivating, the phenotype of articular chondrocytes is often modified [34,79,117,118].

## 6. Cell Culture In Vitro

### 6.1. Chondroprogenitors and Mesenchymal Stem Cells

The population of MSCs is considered to be chondroprogenitors under appropriate stimulation conditions [109]. MSCs isolated from bone marrow (BM-MSCs) are the “gold standard” for their experimental monitoring and cultivation. In vitro culture of the collected bone marrow sample is multiplied in vitro in alpha Minimum Essential Medium Eagle (MEM) culture medium with 10% fetal bovine serum and antibiotics on the culture surface for adherent lines in a 5% CO_2_ atmosphere. In the primoculture of BM-MSCs, the formation of a monolayer of cells usually takes three weeks. MSC-adhered cells have a typical fibroblastoid shape. Aside from bone marrow, MSCs are also often isolated from adipose tissue for autologous therapeutic use [119].

### 6.2. Monolayer Culture of Chondrocytes

Chondrocytes for further in vitro cultivation are enzymatically isolated from articular cartilage with collagenase type II and cultured under appropriate conditions in a 5% CO_2_-enriched atmosphere in growth medium, namely Dulbecco’s Modified Eagle Medium (DMEM) with glucose (1 g/L) combined in a 1:1 ratio with Ham’s F12 nutrient mix medium, with 10% fetal bovine serum (FCS) and antibiotics [120]. The cells multiply in vitro gradually. Articular chondrocytes isolated from higher vertebrates and cultured in a monolayer usually stop producing characteristic ECM molecules [36,121]. After the expansion of the chondrocytes in the monolayer, there is a change in the characteristic phenotype. Chondrocytes dedifferentiate and start producing type I collagen, which is a characteristic of the fibroblastic cell type. Alternatively, the cells transdifferentiate, which is a process of chondrocyte hypertrophy, and change to produce type X collagen [122]. Several authors have found a tendency of chondrocytes to dedifferentiate in vitro, while chondrocytes lose the ability to produce specific type II collagen. In dedifferentiated cells, the production of nonspecific type I collagen for hyaline cartilage was observed [36,121]. The dedifferentiation process is associated with a low density of seeded cells in the primary culture. This fact is closely related to the native morphology of hyaline cartilage containing a lower number of chondrocytes compared with the ECM’s volume of cartilage. The primary culture of chondrocytes seeded at a higher density also maintains a polygonal morphology until after passage, when the cell morphology changes to fibroblast-like and there is an associated change in the gene expression of type I and type III collagen [99]. In practice, it is possible to multiply chondrocytes during several passages and subsequently redifferentiate them under optimal 3D conditions [123].

#### Activation Factors

Several studies have used various additives for chondrocyte culture, insulin-transferrin-selenium (ITS), AA, and TGF-β [124,125,126]. ITS together with fetal serum has been shown to increase chondrocyte proliferation in culture and be a beneficial additive in chondrocyte medium. Chua et al. observed reduced dedifferentiation when using ITS and FBS. The number of chondrocytes increased severalfold compared with the standard culture medium control [124]. The growth factors TGF-β and FGF, which act identically on cells in vitro, also contribute to this effect. Several authors reported the use of TGF-β as a stimulation of chondrocyte redifferentiation in vitro [34,125]. An important factor in chondrocyte culture in vitro is the presence of antioxidants, specifically AA. The authors observed the effect of AA on chondrocytes in vitro under the condition of simulated oxidative stress by the action of hydrogen peroxide (H_2_O_2_). A positive effect of AA on chondrocyte viability was recorded in vitro under conditions of simulated oxidative stress [81].

Another monitored factor during in vitro cultivation is the hypoxia of the environment. Avascular cartilage tissue contains an oxygen gradient ranging from 7% in the superficial zone to 2% in the deep zone. Many authors followed the effect of hypoxia and observed the redifferentiation of chondrocytes and the expression of typical types of collagens of hyaline cartilage. They tested the effect of hypoxia on cells in the range from 2% to 5% pO_2_. Jeyakumar et al. noted after two weeks the redifferentiation of chondrocytes under hypoxic conditions, yielding 4% pO_2_ in a culture medium enriched with platelet-rich plasma (PRP). In the study, it was observed that PRP had a stimulatory effect on the gene expression of the markers COL2A1 and SOX9 [127,128,129]. Jahr et al. applied 2.5% pO_2_ in the cultivation of articular chondrocytes while proving the expression of the chondrogenic markers ACAN and COL2A1 and the reduction of expression of the dedifferentiation markers COL1A1 and COL3A1. Several studies have shown that the hypoxic environment has a stimulating effect on the formation of ECM proteins and maintenance of the production phenotype of articular chondrocytes [127,130]. An essential factor that affects the metabolism of chondrocytes is temperature. Current studies show that a lower temperature during cultivation, namely around 32 °C, also has an effect on reducing the rate of dedifferentiation of chondrocytes by modulating biochemical and metabolic processes in the cells. Hypothermia also affects chondrocytes in vitro by slowing the rate of cell proliferation in both monolayer and pellet cultures [113,117].

### 6.3. Three-Dimensional Culture

The cell morphology of chondrocytes is significantly related to the type of cell culture used. From the previous information, it follows that the phenotypic stability of chondrocytes in culture is conditioned by several factors, including the density of seeded cells. In the case of a “high-density” culture, chondrocytes are able to maintain the original phenotype of collagen II production for the next passage, while the gene expression for COL2 is less stable compared with the proteoglycan ACAN. It appears that the type of in vitro culture influences the state of dedifferentiation and redifferentiation of chondrocytes. Benya et al. described the results of the experiment in a well-known study dealing with articular chondrocytes in an animal experiment. After recording the monolayer culture of chondrocytes and dedifferentiation of their COL1 production phenotype, a spherical shape of chondrocytes and COL2 production was observed after changing the culture conditions in agarose gel [36,127]. It is generally accepted that 3D culture has a beneficial effect on maintenance of the differentiated phenotype and chondrocyte redifferentiation.

The reverse process of redifferentiation of chondrocytes in gel suspension cultures was also demonstrated in vitro. Acquisition of a morphologically round cell shape acts to reduce chondrocyte proliferation in 3D cultures. Currently, there are several specialized types of 3D cultures, such as cultivation in spinner flasks [131,132], in pellets, on 3D solid matrices, or in collagen gels [36], agarose, or alginate [133].

#### 6.3.1. Alginate Culture

Alginate, a natural polysaccharide, has high biocompatibility and considerable biodegradability. The advantage of alginate is in its properties, which have been designed to adapt to the 3D arrangement of cells that mimic the arrangement in hyaline cartilage. Unbranched linear copolymers of β-D-mannuronic acid (M) and bound α-L-guluronic acid (G) form gel alginates [122,134]. Alginate can be isolated from brown seaweed of the class Phaeophyceae or isolated from bacteria such as *Pseudomonas* [122]. The material can be prepared in various forms, such as scaffolds, hydrogels, foams, microspheres, and fibers [135]. Divalent cations Ca^2+^ and Mg^2+^ act on the crosslinking of alginate hydrogels [122]. As a polysaccharide with an anionic nature dependent on pH, it actively interacts with cationic polyelectrolytes and proteoglycans. The most common 3D alginate encapsulation system is prepared by combining an alginate solution with a chondrocyte suspension either through a drip, syringe, automatic pipette, or automatically. Chondrocytes encapsulated in alginate maintain a round cell shape. Bian et al. observed alginate microspheres with encapsulated MSCs and the addition of TGF-β in HA hydrogels. They noted the positive effect of TGF-β encapsulated in alginate on chondrogenic differentiation [136]. A significant benefit of encapsulation of dedifferentiated chondrocytes in alginate microspheres causes reverse redifferentiation and type II collagen production. The results show that alginate improves the process of redifferentiation of chondrocytes [34,137].

Alginate is prepared either in a homogeneous way with a compression modulus from 1 to 1000 kPa or through a shear modulus from 0.02 to 40 kPa [135]. Lee et al. observed the shear properties of alginate applied in a layered manner in combination with animal chondrocytes. They monitored the strength and shear properties of the cell and alginate complex after long-term in vitro culture. After more than 2 months of cultivation, they observed a twofold improved strength and a sixfold increased shear modulus. Overall, they evaluated the most significant increase in shear strength. Furthermore, the authors noted a gradual process of integration that occurs between the layers of alginate [134]. The advantage of alginate is in its malleability and biocompatibility. The material has a reparative potential and can function as a cultivation 3D matrix or a transport system of cells with therapeutics. Alginate is a promising material widely used for 3D culture of chondrocytes [135].

#### 6.3.2. Collagen- and Hyaluronan-Based Scaffolds

As one of the main components of cartilage, collagen is characterized by low antigenicity and biocompatibility. Collagen matrices are used in the form of matrices, fibers, and coatings [138]. The advantage of collagen is its stimulating effect on cell proliferation, adhesiveness, and the production of ECM components [139]. HA ensures hydration and smoothness of the cartilage surface. HA is characterized by its ability to bind fluids in the ECM and thereby improve the compressive strength of the internal environment [140]. In a cell culture, a positive effect of HA on COL2 and proteoglycan production by chondrocytes was noted. Akmal et al. tested a bioink containing collagen and HA and noted increased production of proteoglycans [141]. The combination of collagen with HA or alginate mimics the ECM environment in cartilage. The authors combined COL I with the tyramine derivative of HA to create a bioink that approximates ECM in its properties [140]. Chen et al. tested the combination of collagen with HA in the form of a scaffold for connective tissues. They observed the ability of collagen to structure the material and cell viability [142]. Collagen/HA scaffolds were seeded with MSCs and tested for chondrogenic differentiation. Amann et al. noted increased expression of type II collagen and GAGs after three weeks of culture [143]. It has been confirmed in many studies that the adhesiveness of biomaterials can be improved through collagen coating. Collagen is a promising structural protein suitable for use in chondral tissue repair [138,144].

#### 6.3.3. Gelatin Hydrogel

Polymeric hydrogels have a high water content and are characterized by specific chemical and physical properties. Hydrogels mimic the ECM environment that enables cell adhesion, while this complex acquires mechanical properties characteristic of tissue in the organism. Gelatin hydrogels attempt to simulate the ECM environment in cartilage. Hydrogels are proposed as a biocompatible material suitable for chondrocyte proliferation. In the form of a gel, chondrocytes are able to encapsulate themselves. The stiffness of the substrate has an effect on several cellular events, such as proliferation, differentiation, and apoptosis [145]. The 3D environment promotes mechanical properties to maintain the cell phenotype. In the case of a scaffold, it is necessary to optimize the stiffness of the matrix, which depends on the nature of the cell and the type of simulated tissue. As a 3D culture material, Li et al. used selected concentrations of gelatin methacryloyl (GelMA) macromers according to Young’s modulus in the range from 4 to 30 kPa. They monitored the effect of changing the degree of pressure, and thus the resulting stiffness of the hydrogel, on the phenotype of chondrocytes [146]. Li et al. tested different types of stiffness of GelMA gels. After 14 days of cultivation, the authors evaluated the morphology and production of proteoglycans. The chondrocytes cultured in high-stiffness gels showed a microscopically round typical cell morphology. They already demonstrated a significant production of proteoglycans through histological safranin-O staining compared with chondrocytes cultured in gels with less stiffness. For chondrocytes, they recommended a higher gel stiffness of 30 kPa as being ideal for chondrocyte morphology. High viability for the chondrocytes after 7 days was recorded in the GelMA hydrogels [146,147].

### 6.4. Biocompatible Synthetic Scaffold

Scaffolds are most often used in combination with seeded cells as a suitable material for various applications in tissue engineering. The natural morphology of chondrocytes in the extra environment around them is ensured by 3D cultivation. This cell shape ensures efficient metabolism and the transfer of active biomolecules between cells.

In vitro testing of various conditions and factors, such as the porosity and ratio of in-dividual components of the material, is possible The properties that the scaffold provides for the organism are the biocompatibility, biological degradability of the material, pore-forming structure, and suitable mechanical properties. The advantage of 3D cultivation is keeping the cell in a spherical shape [148].

Rogan et al. used two types of hydrogel systems to simulate chondrocyte differentiation and ECM mimicry: synthetic polyethylene glycol (PEG) and bioactive hydrogel, a complex containing parts of PEG and CS hydrogels [149]. Vergese et al. used a photo polymerization process to encapsulate the MSCs while the cell distribution was homogenous. The photopolymerization process was assessed to be an effective method with minimal cytotoxicity. In the hydrogels with CS and PEG, self-aggregating clusters of cells with cartilage ECM production were observed for 3 weeks. Self-regulation was not observed in the PEG scaffolds, while the type of scaffold had no effect on the cell morphology. The cells were morphologically typically round. From the results of differentiation of MSCs after the end of cultivation, the presence of a higher expression of genes typical for chondrocytes on the CS/PEG scaffold was proven compared with the PEG hydrogel scaffold. The study revealed the stimulating effect of CS and the suitability of the tested synthetic material PEG for the differentiation of chondrocytes from MSCs [150].

A current topic is the testing of synthetic polymers of biocompatible materials based on polylactic acid (PLA), polyhydroxybutyrate (PHB), and thermoplastic starch (TPS) with improved mechanical properties suitable to be scaffolds for cell engraftment. Balogová et al. described a biocompatible material consisting of PLA polymers with the addition of ceramics suitable for 3D cultivation of chondrocytes in which long-term biodegradation was monitored in a simulated physiological environment. The results of the study showed that the biodegradation of the material was gradual, and the material appeared to be stable in the organism’s environment [151]. Trebuňová et al. tested the composite material PLA/PHB for cultivation and cytotoxicity while evaluating the most optimal ratio of individual components for the mentioned parameters. They observed that the composite material did not show any significant signs of cytotoxicity [152].

### 6.5. 3D Printing Technology

There is an ideal application in regenerative medicine for 3D printing, and it is an advanced method for personalized cartilage defect repair. The advantage of 3D printing is the possibility of precisely defining and creating the complex structure of the resulting material, imitating the layers in the tissue. The process is programmed and controlled by a computer using specialized software. It is possible to deposit materials in layers according to CAD and precisely adapt to the irregular shape of the defect [153]. There are two known printing approaches: acellular processes and a new method, namely bioprinting with incorporated cells. Currently, several types of 3D printers have been developed that can be used for bioprinting: inkjet printing, laser 3D printing, and bioextrusion. In the bioextrusion process, bioinks such as alginate, gelatin, HA, and the synthetic polymers poly(ethylene)-glycol (PEG), polycaprolactone (PLC), and polyglycolic acid (PGA) are used in combination with cells [154]. Biocompatibility, appropriate porosity, and degradability are the necessary properties of biomaterials used for regeneration purposes. Optimal biomechanical parameters for the material used are also essential. The used material must functionally come as close as possible to the tissue it imitates. The mentioned materials can be combined in different proportions and as a result form a bioink suitable for use with cells to create a reparative cartilage defect implant.

As for the application of the cells, the most promising method for the repair of hyaline cartilage defects appears to be the use of autologous chondrocytes isolated from the hyaline cartilage of the non-load-bearing surfaces of the articular cartilage of the knee joint. MSCs would be obtained from various sources of the body, most often bone marrow and adipose tissue [53]. Hyaline cartilage is found on several body structures, such as articular surfaces of the bones, ribs, and nose [16]. Extrusion-based bioprinting (EBB) is used for bioprinting hydrogels in combination with cells in the selected ink. Currently, bioinks in combination with chondrocytes based on alginate, gelatin, a decellularized extracellular matrix (dECM), and HA have been studied [155,156]. The most complicated part of 3D printing technology is the construction of the tissue with the corresponding layers in the articular cartilage. The EBB system is suitable for forming the layered organization of articular cartilage. Tang et al. used gelatin methacrylate (GelMA) and PLA with seeded chondrocytes to reconstruct the auricle. They assessed the printing accuracy, biomechanics, and chondrocyte viability. GelMA ensured the fixation of cells and distribution in PLA scaffolds. After two months of cell seeding, chondrocyte proliferation and type II collagen formation were observed [157]. Overall, 3D printing with a combination of materials and cells represents a unique perspective in regenerative medicine [153]. The clinical use and therapy for cartilage defects from 3D printed matrices seeded with chondrocytes is still not fully resolved. The question of sufficient strength and adhesion of the material in the place of the defect is complicated.

### 6.6. Cartilage Tissue Engineering

Cytotherapy is a promising direction in the advanced regeneration of damaged articular cartilage in the most common diseases: OA and RA. Tissue engineering focuses on a comprehensive solution for the repair of articular cartilage defects. Solutions for the most accurate imitation of ECM cartilage are approached using natural gel materials based on proteins or polysaccharides, solid synthetic scaffolds, or a combination thereof. The biomaterial must meet the properties suitable for cells and its use for in vivo cartilage repair. The essential properties are biocompatibility, cytotoxicity, and surface treatment with adaptation for cell adhesion [158] (Figure 3). An important property is the gradual degradability of the biomaterial, which enables tissue remodeling and reconstruction in vivo [104]. There are commercial products that are certified and suitable for the repair of articular cartilage defects. Chondro-Gide^®^ is a collagen membrane designed for cartilage regeneration. It is prepared from xenogenic pig collagen, where its properties support the regeneration of cartilage. The membrane is characterized by a smooth surface with a compact structure and a rough and porous inner layer. Method matrix-induced chondrocyte autologous implantation (MACI) combines the use of cells and a collagen membrane with Tisseel^®^ fibrin glue [159]. The collagen type I/III scaffold is embedded with autologous chondrocytes, and the fibrin glue ensures attachment at the implantation site. Zheng et al. evaluated in detail the repair of patients’ defects after MACI therapy via scanning electron microscopy while observing an organized two-layer membrane formed by collagen fibers with attached round-shaped chondrocytes. The chondrocyte phenotype was also confirmed by the expression of type II collagen. Hyalograft^®^C is an HA-based scaffold and allows the implantation of expanded autologous chondrocytes in 3D form [159]. The method of repairing the defect is possible without covering the defect with a periosteal flap. Pavesio reported that the use of Hyalograft^®^C in combination with autologous chondrocytes is an effective method of therapy for articular cartilage lesions [160]. Therapy of defects through atelocollagen gel in combination with autologous chondrocytes is also promising [161]. Tohyama et al. monitored the outcomes after repair of knee cartilage defects after a two-year interval in a multicenter study. From the clinical arthroscopic study, as a result of the repair of defects with atelocollagen with cells, they noted an improvement in the condition of the patients by monitoring the Lysholm score, and the repair condition was evaluated in part of the patients to be improved with minimal adverse effects [162]. Scaffolds based on collagen fibers or HA provide a spatial arrangement for adhered cells that are capable of proliferation and synthesis of cartilage ECM molecules [163].

## 7. Clinical Approach in Advanced Cartilage Therapy

OA and RA are manifested by the limitation of joint movement, which is accompanied by frequent pain [164]. The most radical solution for the most severe form of OA is total arthroplasty. Repair techniques such as the microfracture technique or mosaicplasty are used for defects of smaller proportions. Awls that have been specially designed for the microfracture technique are used to create multiple perforations. The disadvantage of the mentioned repair procedures is the formation of incomplete fibrocartilaginous cartilaginous tissue [123,165]. The technique of autologous chondrocyte implantation (ACI) is based on the autologous approach and application of the patient’s own multiplied cells. Continuously, this method is improved by progress in implantation techniques. ACI can be used in the therapy of osteochondral defects (>8 mm^2^). Brittberg et al. were the first to publish a procedure for the therapy of chondral defects by means of ACI in 1994 [166]. In their autologous chondrocyte implantation procedure, they used the periosteum as a membrane covering the defect. An unfavorable fact of the therapy is the presence of hypertrophy in the place of the repaired defect in some patients, which was related to the use of periosteum. Currently, collagen membranes are used for the overlay, with a more favorable final effect. The most current technique, modified by ACI, is the use of a membrane in combination with cells, namely MACI. In the MACI method, a multiplied population of autologous chondrocytes is applied to a biodegradable 3D scaffold that contains collagen I and III or HA. The 3D construct with cells is transferred into the defect and subsequently stabilized using fibrin glue. As a result of defect therapy (≥3 cm^2^) through MACI, the results after a 5 year period were clinically more favorable compared with the microfracture repair technique [167,168]. MACI is an advantageous technique with a simpler surgical procedure and is also a more reproducible method. The long-term results and benefits for patients have been compared from the previous studies. A 2020 study evaluated and followed 1000 adult patients who underwent MACI. The preliminary results have been published, and only less than 3% of the patients had adverse effects [159]. The defect on the patella was repaired most often, followed by repair of the medial condyle of the femur [169,170,171,172]. Many authors confirmed the fact that isolated and cultured chondrocytes change their physiological properties and phenotypic expression in vitro, which is a serious problem in monolayer culture and in the use of chondrocytes in tissue engineering [36,121].

## 8. Conclusions

Currently, there is rapid growth in multidisciplinary medical bioengineering, and new methods with therapeutic application in orthopedics are being developed. The interaction between articular cartilage cells, chondrocytes, and the scaffold, which creates an environment that mimics the cartilage ECM, is essential for the construction of the most ideal solution. The development of bioengineering is related to the possibility of culturing and multiplying cells in vitro. It is possible to prepare a transplantable graft from a minimal tissue fragment. There is a phenotype of chondrocytes in articular cartilage with type II collagen, and ACAN production is an important condition for maintaining the biological and mechanical properties of cells and their function in the musculoskeletal system [36,173]. In order to create artificial cartilage tissue, it is necessary to study the complex relationships between the phenotype of chondrocytes, ECM molecules, and growth factors [146]. Several types of cells are suitable for in vitro cultivation, namely articular chondrocytes, MSCs from various sources, and pluripotent stem cells that are capable of differentiation into cartilage tissue. Autologous chondrocytes are the most widely used and approved cell type for chondral repair. A disadvantage of chondrocyte culture is the limited number of cells during isolation and possible dedifferentiation during expansion. The present studies confirmed that 3D culture in vitro is required to maintain the type II collagen production phenotype of articular chondrocytes [119]. Stem cells have a crucial role in the repair of tissues in an organism. Adult stem cells with the ability of self-renewal and optimal replicative potential can be used in regenerative bioengineering procedures [174]. Thus far, many types of 3D in vitro culture in natural gel materials such as alginate, collagen, and agar have been described. These are ideal for maintaining the round shape of articular chondrocytes and providing similar properties to the ECM in the organism. It is generally accepted that 3D culture has a beneficial effect on the maintenance of the differentiated phenotype and redifferentiation of chondrocytes.

When using synthetic scaffolds, an advantageous feature is the sufficient strength of the material during culture, as well as the provision of strength parameters for the future graft when repairing defects [148]. In vitro, the scaffold is most often used for tissue engineering testing. The properties that the scaffold provides for the organism are biocompatibility, biological degradability of the material, a pore-forming structure, and suitable mechanical properties. For the repair of articular cartilage defects, various methods are being developed and improved using bioengineering, where the cooperation of several factors, such as biological and mechanical factors, is necessary (Table 2).

## Figures and Tables

**Figure 1 ijms-24-17096-f001:**
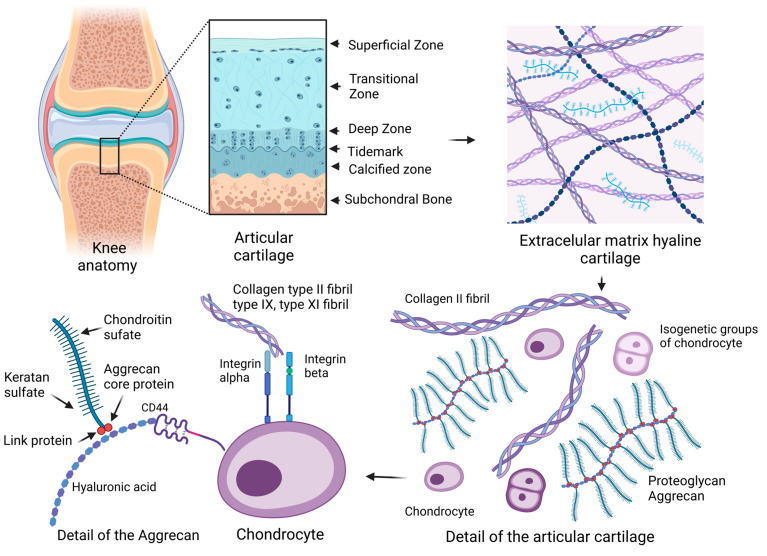
Schematic of an articular joint and details of the extracellular matrix (ECM) and chondrocytes. The arrows between the images mean a more detailed view in the overall scheme. Hyaline cartilage is composed of a complex structured ECM with the presence of collagen type II (COL2) and the proteoglycan aggrecan. Articular cartilage affects collagen fiber orientation, cell density, and cytomorphology, which creates a hierarchically organized cartilage structure. Glycosaminoglycans (GAGs) and aggrecans (ACANs) are present in the ECM, which contain chains of chondroitin sulfate (CS) and keratan sulfate (KS). ACANs, as the main proteoglycans of hyaline cartilage, interact with hyaluronan (HA) fibers and form complex proteoglycan aggregates. ACANs are bound to the HA fiber by link and core proteins. Mature chondrocytes of hyaline cartilage are situated in lacunae in isogenetic groups of several cells in a territorial matrix located in hyaline cartilage in a specific physicochemical environment. The membrane receptor of chondrocytes is the hyaluronic receptor CD44. Articular chondrocytes express alpha and beta integrins of several types, which interact with COL2 and proteoglycans molecules in the ECM (created with BioRender.com, accessed on 1 November 2023).

**Figure 2 ijms-24-17096-f002:**
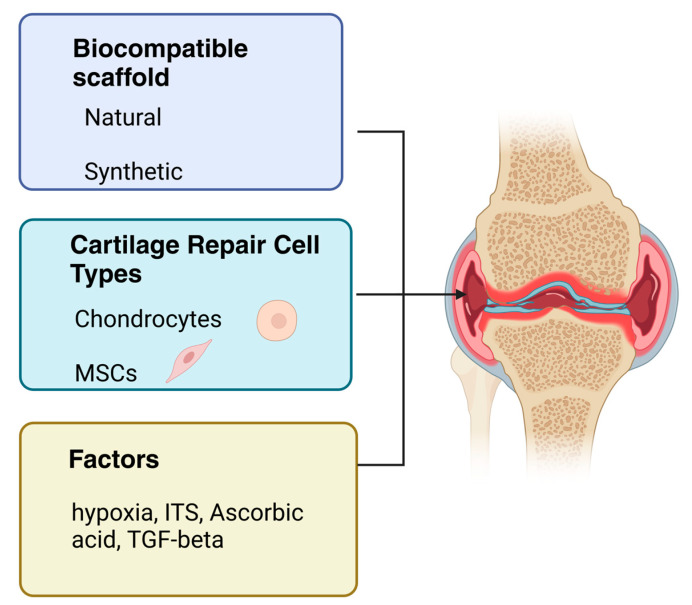
Coordinating components in the repair of articular cartilage. The challenging task of tissue engineering is to prepare an optimal solution to the problem of repairing chondral defects. The repair of articular cartilage is a process in which the main factors are the type of applied cells, the type of scaffold on a natural or synthetic basis, and growth factors for the successful repair of chondral defects. Mesenchymal stem cells (MSCs) are frequently used cells in the broad field of tissue engineering. MSCs participate in regeneration processes, angiogenesis, and differentiation in the organism. Chondrocytes for in vitro multiplication can be isolated from the non-load-bearing area of articular cartilage. The dedifferentiation process of chondrocytes was also observed in vitro. The primary culture of chondrocytes seeded at a higher density maintains a polygonal morphology and is suitable for initial cell multiplication. Further in vitro cell culture is appropriate in a three-dimensional environment to maintain the natural phenotype of collagen type II-producing chondrocytes (created with BioRender.com, accessed on 1 November 2023).

**Figure 3 ijms-24-17096-f003:**
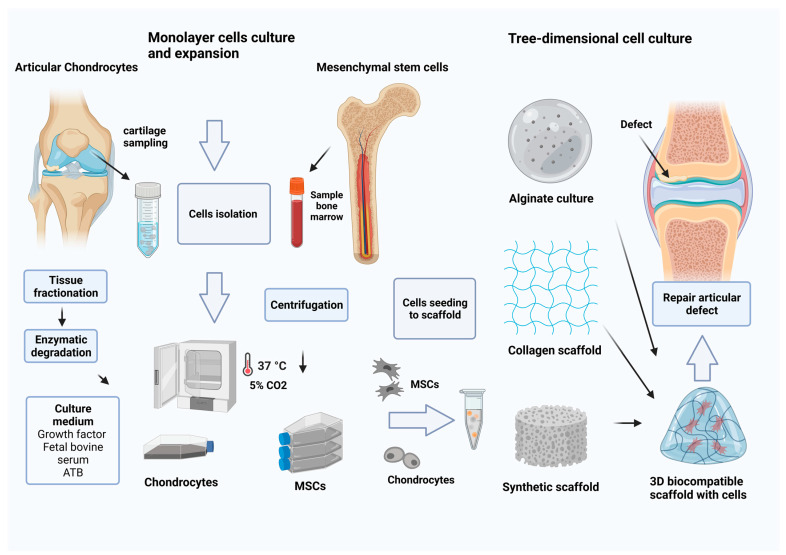
The procedure for the preparation of 3D scaffolds with seeded cells. The arrows between the images indicate the direction of the process of preparing the 3D cell-seeded scaffold. Two cell isolation options: chondrocytes or mesenchymal stem cells (MSCs). Chondrocytes can be isolated from articular cartilage. The tissue is enzymatically digested, and the chondrocyte suspension is isolated. The second option is the isolation of MSCs from bone marrow or adipose tissue with subsequent in vitro cultivation in incubator conditions at 37 °C with 5% CO_2_. The cells multiply in a medium containing growth factors and antibiotics. MSCs and chondrocytes are suitable for implantation of a natural or synthetic scaffolds. Scaffolds are most often used in combination with seeded cells as a suitable material for various applications in tissue engineering, and 3D cultivation ensures the natural morphology of chondrocytes in the extra environment around them. Alginate and collagen have been shown to be very suitable candidates due to their biocompatibility, tunable mechanical properties, and ability to provide structural support for MSCs. Alginate scaffolds provide an environment conducive to chondrogenesis and facilitate the differentiation of MSCs into chondrocyte-like cells. They can be tailored by incorporating growth factors, bioactive molecules, and mechanical stimuli to enhance cartilage regeneration. Collagen scaffolds mimic the natural extracellular matrix (ECM), supporting cell attachment, migration, and proliferation. Encapsulated MSCs in collagen scaffolds show increased chondrogenic differentiation potential, with the scaffold acting as a structural support for neocartilage formation. The use of synthetic scaffolds in combination with MSCs offers several advantages in cartilage repair. Scaffolds provide a stable and organized structure, preventing the formation of fibrocartilage or scar tissue inferior to natural hyaline cartilage. They can be tailored to meet the specific needs of the patient, ensuring optimal fit and integration with the surrounding tissues (created with BioRender.com, 1 November 2023).

**Table 1 ijms-24-17096-t001:** Markers for hyaline cartilage chondrocytes, hypertrophic chondrocytes, and fibrocartilage [8,91,93].

Hyaline Cartilage	Hypertrophic Cartilage	Fibrocartilage
SOX9, SOX5, SOX6	COL10	COL1
COL2, COL9, COL11	MEF2C	COL3
ACAN	RUNX2	α-Smooth muscle actin (α-SMA)
RUNX1	RUNX3	Fibroblast-specific protein 1 (FSP1/S100A4)
Small novel rich in cartilage (SNORC)	-	TIMP1, Cell migration-inducing protein (CEMIP)
WW domain-containing protein 2 (WWP2), MicroRNA-140 (miR-140)	-	-

**Table 2 ijms-24-17096-t002:** Abbreviations used often in the text.

Abbreviation Glossary	Acronym
Advanced glycation end products	AGE
Aggrecan	ACAN
Ascorbic acid	AA
Autologous chondrocyte implantation	ACI
Bone morphogenetic protein	BMP
Chondroitin sulfate	CS
Collagen type I	COL1
Collagen type II	COL2
Collagen type II alpha-1 gene	Col2A1
Drosophila mothers against decapentaplegic proteins	Smad
Extracellular matrix	ECM
Fibroblast growth factor	FGF
Gelatin methacryloyl	GelMA
Glycosaminoglycans	GAGs
High-mobility group box	HMGB
Intercellular adhesion molecule 1	ICAM-1 (CD54)
Keratan sulfate	KS
Matrix metalloproteinases	MMPs
Matrix-induced autologous chondrocyte implantation	MACI
Mesenchymal stem cells	MSCs
Mitogen-activated protein kinases	MAPK
Osteoarthritis	OA
Synthetic polyethylene glycol	PEG
Polycaprolactone	PLC
Polyglycolic acid	PGA
Proteoglycan-4	PRG4
Reactive oxygen species	ROS
Retinoid acid	RA
Retinoid acid receptor	RAR
Runt-related transcription factor 1	RUNX11
Small Novel Rich in Cartilage	Snorc
SRY-Box Transcription Factor	SOX
Tissue metallopeptidase inhibitor 1	TIMP1
Toll-like receptor	TLR
Transforming growth factor beta	TGF-β2
Vascular cell adhesion molecule 1	VCAM-1 (CD106)
Vascular endothelial growth factor	VEGF

## Data Availability

Not applicable.

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
