# Peer review of "Human Chondrocytes, Metabolism of Articular Cartilage, and Strategies for Application to Tissue Engineering"

_ijms, 2023, doi:10.3390/ijms242317096_

Round 1

Reviewer 1 Report

Comments and Suggestions for Authors

Bačenková et al. summarized the current understandings of cell culture methods and tissue engineering techniques that can lead to cartilage regeneration. Although this review article is massive in length, this reviewer believes that the manuscript requires some revisions prior to publication. Specifically, there are issues with the misuse of reference articles, and the use of graphical content would improve the contents of manuscript.

1.      The manuscript contains only two figures that are focused on the structure of cartilage tissue. Considering that the cell culture methodologies and tissue engineering techniques are major topics related to the tissue engineering in this manuscript, graphical elements such as additional figures or tables are required to increase readers’ understandings.

2.      Suggestions on the content of manuscript:

In line 104, CS is a major component of sulfated GAG comprising the cartilage. Therefore, introduction to CS should be added on this paragraph.

In line 271-273, there are many diseases that occurs in the articular cartilage, other than OA and RA.

In line 534, the section title should include type collagen II, considering the content.

In line 627-628, this sentence should contain examples for hyaline cartilage, considering the title of this manuscript.

In line 627-642, this part should contain more example for the application of these technologies in OA or RA, diseases occurring in the articular cartilage, as covered in section 4.

3.      Citation of original research articles: This manuscript inappropriately used many review articles instead of original research articles. These include:

In line 176-177, reference 34 at the end of this sentence.

In line 549-550, reference 102 is a review article.

4.      There are many parts that are missing the reference article for their statement. The parts that requires addition of supporting evidence include:

Line 170-177, each sentence requires reference article separately, where appropriate.

Line 184-194, each sentence requires reference article separately, where appropriate.

Line 195-203, each sentence requires reference article separately, where appropriate.

Line 222-229, each sentence requires reference article separately, where appropriate. In addition, original research article is preferred.

Line 229-230, this sentence requires reference article and seems to be in out of context.

Line 237-245, each sentence requires reference article separately, where appropriate.

Section 3.2, each sentence requires reference article separately, where appropriate. There are so many unreferred sentences.

Line 278-283, each sentence requires reference article separately, where appropriate. In addition, original research article is preferred.

Line 285-294, each sentence requires reference article separately, where appropriate.

Line 303-305, this part requires reference articles.

Line 319-329, each sentence requires reference article separately, where appropriate.

Line 335-343, each sentence requires reference article separately, where appropriate.

Line 343-358, each sentence requires reference article separately, where appropriate. In addition, the only cited reference 61 does not cover the whole content over this range.

Line 465-473, each sentence requires reference article separately, where appropriate.

Line 476-478, several studies should come here, as written.

Line 485-496, each sentence requires reference article separately, where appropriate. Notably, the only cited reference 37 was published in 1982 and seems somewhat outdated.

Line 562-575, each sentence requires reference article separately, where appropriate.

Section 6.4, there are only very few references cited in this paragraph. More references related to recent findings in biocompatible synthetic scaffold should be added.

Line 610-626, each sentence requires reference article separately, where appropriate.

Line 658-659, this part requires reference article, at the end of “final effect”.

5.      Inappropriate citations that should be replaced include:

In line 67, regarding reference 11, ear is composed of elastic cartilage, not hyaline cartilage.

In line 78, regarding reference 14, disc is majorly composed of fibrocartilage, not hyaline cartilage.

In line 94, the reference 15 is not related to the content of the sentence.

In line 104, the reference 18 and 19 are not related to the content of the sentence.

In line 122, the reference 20 is not related to the content of the sentence.

In line 155, the reference 29 is not related to the content of the sentence.

In line 182, the reference 37 is not related to the content of the sentence.

In line 300, the reference 53 is not related to the content of the sentence.

In line 313, the reference 55 is not related to the content of the sentence.

In line 322, the reference 10 is not related to the content of the sentence.

In line 502, the reference 28 is not related to the content of the sentence.

In line 533, the reference 96 is not related to the content of the sentence.

In line 543, the reference 62 is a book and this did not “test” such hypothesis. Original research article should come here.

In line 568, the reference 106 is not related to the content of the sentence.

In line 638, the reference 117 is not related to the content of the sentence.

6.      Suggestions regarding expressions:

In line 11, what are “Production cells”?

In line 18-20, the last sentence is incompletely structured.

In line 57, is the proportion based on dry weight?

In line 88-104, “s” at the end of ACANs has to be used only when necessary.

In line 172-174, this sentence is written unnaturally.

In line 187, TGF-beta is not a receptor.

In line 380 and others, “primoculture” should be replaced to “primary culture”.

In line 427-428, this sentence is written unnaturally.

In line 581, “various variable conditions”?

In line 592-593, two sentences seem to be unnaturally separated.

In line 607-608, this sentence is written unnaturally.

In line 609, “D” should be “3D”.

7.      The abbreviations must appear only once they are introduced for the first time.

Comments on the Quality of English Language

The manuscript is overall well-written.

Author Response

Dear reviewer,

Thank you very much for thorough review of our manuscript.

Reviewer 2 Report

Comments and Suggestions for Authors

Bačenková et al. present a review on the basic biochemical processes in hyaline cartilage based on structural amino acids and current cell populations. The overview also discussed the possibilities of in vitro culture of chondrocytes and detailed descriptionsof current methods of three-dimensional (3D) cultivation. While the manuscript presents exciting data there are a few issues to be addressed.

(1)   The title does not align with the content of the manuscript. The manuscript's title places emphasis on chondrocytes, yet there is extensive content discussing other aspects that may require adjustment for better alignment.

(2)    The author should give more related applications to tissue engineering on chondrocytes.

(3)    What does Table 2 signify in line 329, and what references are associated with it [29, 59]? Please carefully review and revise Table 2.

(4)    As a review, there are some points needed to be involved e.g., the effect of mechanics on the chondrocyte, articular cartilage, and their activity…, in the MS.

Comments on the Quality of English Language

Minor editing of English language required.

Author Response

Thank you very much for a thorough review of our manuscript.

Round 2

Reviewer 2 Report

Comments and Suggestions for Authors

I think the present form is acceptable. Thus, there is no additional comment.